



**1  Technical Note:**

**2  A two-sided affine power scaling relationship to represent the**

**3  concentration–discharge relationship**

José Manuel Tunqui Neira[1,2], Vazken Andréassian[1], Gaëlle Tallec[1] & Jean-Marie Mouchel[2]
[(1)] Irstea, HYCAR Research Unit, Antony, France
[(2)] Sorbonne Université, CNRS, EPHE, UMR Metis 7619, Paris, France

**7  Abstract**

This technical note deals with the mathematical representation of concentration–discharge
relationships. We propose a two-sided affine power scaling relationship (2S-APS) as an alternative to
the classic one-sided power scaling relationship (commonly known as "power law"). We also discuss
the identification of the parameters of the proposed relationship, using an appropriate numerical
criterion. The application of 2S-APS to the high-frequency chemical time series of the Orgeval-Oracle
observatory is presented (in calibration and validation mode): It yields better results for several
solutes and for electrical conductivity in comparison with the power law relationship.

**15  Keywords**

Concentration–discharge relationships; log–log space; power law, high-frequency chemical data

**17  1. Introduction**

The relationship between solute concentrations and river discharge (from now on "C-Q relationship")
is an age-old topic in hydrology (see among others Durum, 1953; Hem, 1948; Lenz and Sawyer, 1944).
It would be impossible to list here all the articles that have addressed this subject, and we refer our
readers to the most recent reviews (e.g., Bieroza et al., 2018; Botter et al., 2019; Moatar et al., 2017)
for an updated view of the ongoing research on C-Q relationships.
Many complex models have been proposed to represent C-Q relationships, from the tracer mass
balance (e.g., Minaudo et al., 2019) to the multiple regression methods (e.g., Hirsch et al., 2010).
Nonetheless, for the past 50 years the simple mathematical formalism known as "power law" has
enjoyed lasting popularity among hydrologists and hydrochemists (see, e.g., Edwards, 1973;
Gunnerson, 1967; Hall, 1970, 1971). Over the years, however, some shortcomings of this relationship
have become apparent: Recently, Minaudo et al. (2019) mentioned that, "fitting a single linear
regression on C-Q plots is sometimes questionable due to large dispersion in C-Q plots (even log



transformed)". Also, Moatar et al. (2017) present an extensive typology of shapes (in log–log space)
for the French national water quality database, which shows that the power law must be modified to
represent the C-Q relationship for dissolved components as well as for particulate-bound elements.
This technical note presents a two-sided affine power scaling relationship (named "2S-APS") that can
be seen as a generalization of the power law. And although we do not wish to claim that it can be
universally applicable, we argue here that it allows for a better description and modeling of the C-Q
relationship of some solutes as a natural extension of the power law.

## 2. Test dataset

We used the half-hourly (every 30 min) hydrochemical dataset collected by the in situ *River Lab*
laboratory at the Oracle-Orgeval observatory (Floury et al., 2017; Tallec et al., 2015). A short
description of the study site is given in Appendix 1. We used dissolved concentrations of three ions –
sodium [$Na^+$], sulfate [$S-SO_4^{2-}$], and chloride [$Cl^-$] – as well as electrical conductivity (EC). This dataset
was collected from June 2015 to March 2018, averaging 20,700 measurement points.
As our main objective in this note is to compare the performance of two relationships (the new 2S-
APS and the classic power law), we divided our dataset into two parts to perform a split-sample test
(Klemeš, 1986): We used June 2015 to July 2017 for calibration (of both relationships), and August
2017 to March 2018 for validation. Table 1 presents the main characteristics of both periods.


**Table 1: Summary of high-frequency dissolved concentrations and electrical conductivity (EC;**
**average, minimum, maximum values and ratio between quantiles 90 and 10 divided by the mean)**
**from the River Lab at the Oracle-Orgeval observatory, divided into two groups: June 2015 to July**
**2017 (calibration period) and August 2017 to March 2018 (validation period).**

| Solute | Unit | Calibration period (June 2015 to July 2017) | | | |
|---|---|---|---|---|---|
| | | *Mean ($\mu$)* | *Min* | *Max* | *$(q_{90}-q_{10})/\mu$* |
| Sodium | mg.L$^{-1}$ | 13 | 2 | 17 | 0.22 |
| Sulfate | Smg.L$^{-1}$ | 19 | 2 | 32 | 0.44 |
| Chloride | mg.L$^{-1}$ | 30 | 4 | 40 | 0.28 |
| EC | µS.cm$^{-1}$ | 704 | 267 | 1015 | 0.23 |
| | | Validation period (August 2017 to March 2018) | | | |
| Sodium | mg.L$^{-1}$ | 13 | 3 | 17 | 0.59 |
| Sulfate | Smg.L$^{-1}$ | 18 | 3 | 26 | 0.70 |
| Chloride | mg.L$^{-1}$ | 29 | 4 | 40 | 0.71 |
| EC | µS.cm$^{-1}$ | 576 | 171 | 813 | 0.65 |

## 51  3. Mathematical formulations

### 52  3.1  Classic one-sided power scaling relationship (power law)

Since at least 50 years ago, a one-sided power scaling relationship (commonly known as power law)
has been used to represent and model the relationship between solute concentration ($C$) and
discharge ($Q$) (Eq. (1)).

$$C = aQ^b \qquad \text{Eq. (1)}$$

From a numerical point of view, the relationship presented in Eq. (1) is generally adjusted by first
transforming the dependent ($C$) and independent ($Q$) variables using a logarithmic transformation,
and then adjusting a linear model (Eq. (2)).

$$\ln(C) = \ln(a) + b.\ln(Q) \qquad \text{Eq. (2)}$$

Graphically, this is equivalent to plotting concentration and discharge in a log–log space, where
parameters $a$ and $b$ can be identified either graphically or numerically, under the assumptions of
linear regression.

### 62  3.2  Limits of the power law

In many cases, the power law appears visually adequate (and conceptually simple), which explains its
lasting popularity. With the advent of high-frequency measuring devices in recent years, the size of



the datasets has exploded, and the C-Q relationship can now be analyzed on a wider span (Kirchner
et al., 2004). Figure 1 shows an example from our own high-frequency dataset: the 17,500 data
points (which correspond to the calibration period of Table 1) represent half-hourly measurements
collected over a 2-year period, during which the catchment was exposed to a variety of high- and
low-flow events, thus providing a great opportunity for exploring the shape of the C-Q relationship.
This being said, we do not wish to imply that a similar behavior could not been identified in medium-
and low-frequency datasets, which remain essential tools with which to analyze and understand
long-term hydrochemical processes (e.g., Godsey et al., 2009; Moatar et al., 2017).

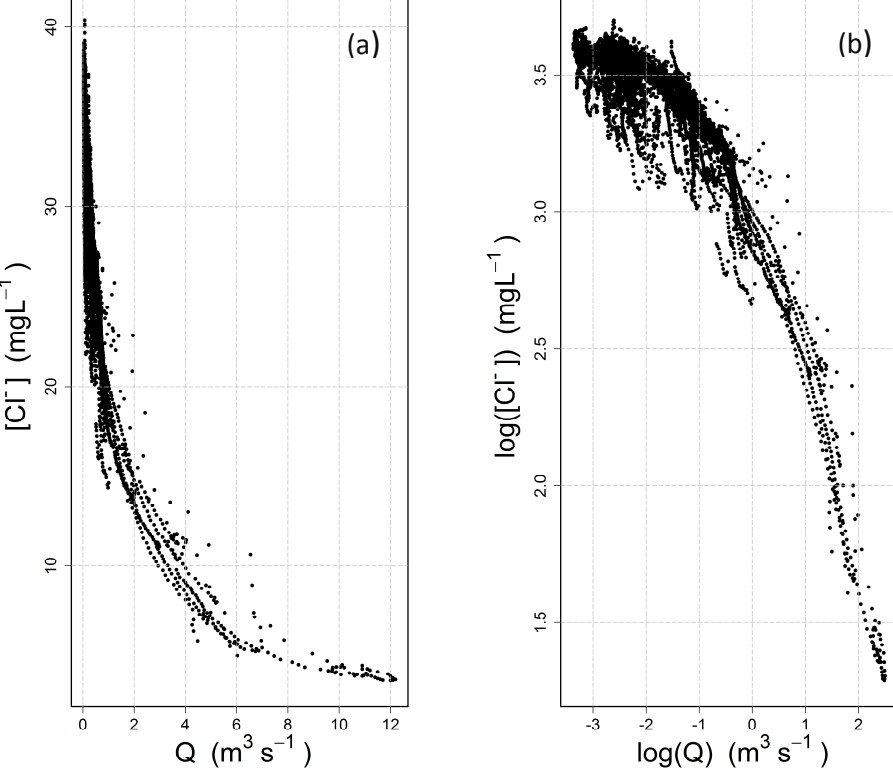


**Figure 1: Concentration–discharge relationship observed at the Oracle-Orgeval observatory**
**(measurements from the River Lab) for chloride ions [Cl⁻]: (a) standard axes, (b) logarithmic axes.**

Figure 1 illustrates the inadequateness of the power law for this dataset: The C-Q relationship
evolves from a well-defined concave shape on the left to a slightly convex shape on the right in the
log–log space. From the point of view of a modeler wishing to adjust a linear model, one has gone
beyond the straight shape that was aimed at. Note that this is true for our dataset, and that it does





not need to always be the case: The log–log space can be well adapted in some situations (see
examples in the paper by Moatar et al., 2017).

### 3.3 A two-sided affine power scaling relationship as a progressive alternative to the power law

As a progressive alternative to the one-sided power scaling relationship (power law), we propose to
use a two-sided affine power scaling (2S-APS) relationship as shown in Eq. (3) (Box and Cox, 1964;
Howarth and Earle, 1979).

$$C^{\frac{1}{n}} = a + bQ^{\frac{1}{n}}$$ Eq. (3)

From a numerical point of view, the relationship presented in **Eq. (3**) is equivalent to first
transforming the dependent ($C$) and independent ($Q$) variables using a so-called Box–Cox
transformation (Box and Cox, 1964), and then adjusting a linear model. In comparison with the
logarithmic transformation, the additional degree of freedom offered by $n$ allows for a range of
transformations, from the untransformed variable ($n = 1$) to the logarithmic transformation ($n \rightarrow \infty$).
This "progressive" property was underlined long ago by Box and Cox (1964): When $n$ takes high
values, Eq. (**3**) converges toward the one-sided power scaling relationship (power law) (Eq. (1)). The
reason is simple:

$$C^{\frac{1}{n}} = e^{\frac{1}{n}lnC} \approx 1 + \frac{1}{n}lnC \text{ when } n \text{ is large.}$$

Thus, for large values of $n$, Eq. (3) can be written as:

$$1 + \frac{1}{n}lnC \approx a + b + \frac{b}{n}lnQ$$

That is equivalent to:

$$\ln C \approx A + b.\ln Q \text{ (with } A = n(a + b - 1))$$

The progressive behavior and the convergence toward the log–log space are clearly evident in Figure
100  2.




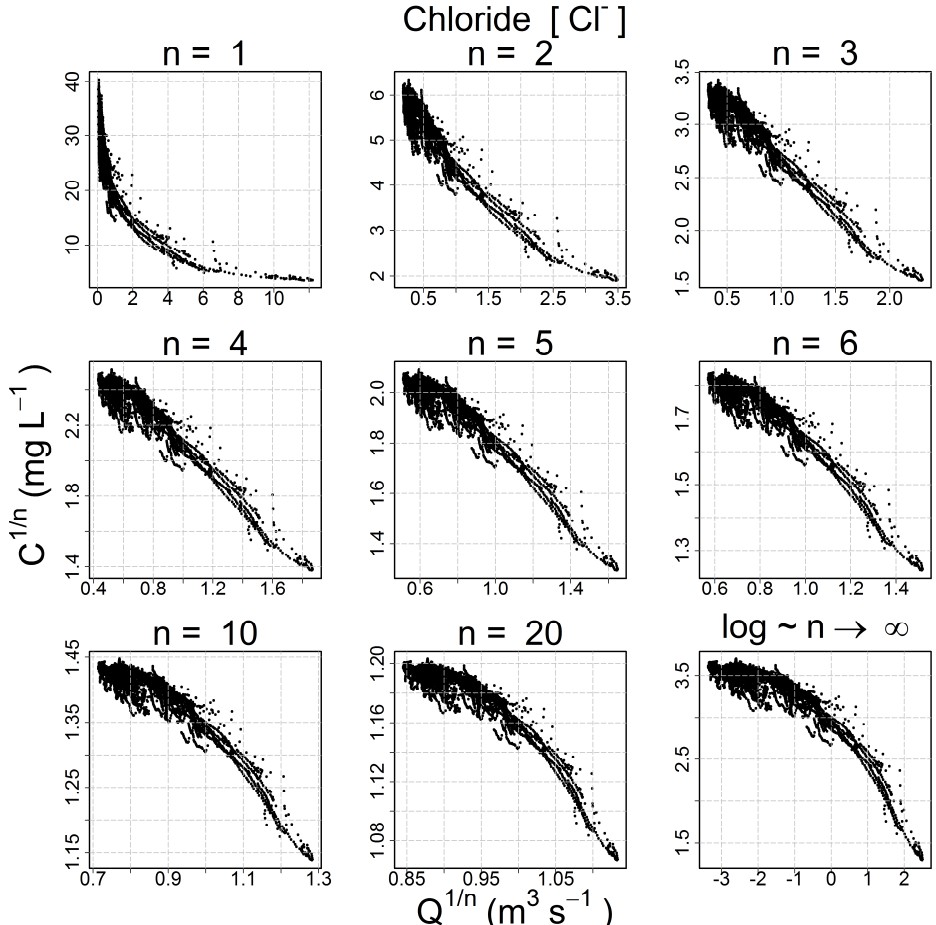

**Figure 2: Evolution of the shape of the concentration–discharge scatterplot for chloride ion with two-sided affine power scaling (2S-APS) and an increasing value of parameter *n*.**

### 3.4 Choosing an appropriate transformation for different ion species (calibration mode)

Because the hydro-biogeochemical processes that control the transport and reaction of ions are different, different ionic species may have a C-Q relationship of distinct shape (Moatar et al., 2017). In Figure 3, we show the behavior of three ions and the EC from the same catchment and the same dataset (all four from the Oracle-Orgeval observatory) with different transformations (*n* =1, 3, 5 and logarithmic transformation). The optimal shape was chosen numerically: We transformed our data series of $C$ and $Q$ using different values of $n$ (i.e., $C^* = C^{1/n}$ and $Q^* = Q^{1/n}$) and logarithmic transformation (i.e., $C^{**} = \log(C)$ and $Q^{**} = \log(Q)$). With these transformed values, we performed





a linear regression and computed parameter $a$ and $b$ and the coefficient of determination ($R^2$) (see
Table 2). The $n$ considered as optimal has the highest $R^2$ value (see Table 2). However, we could also
have followed the advice of Box et al. (2016, p. 331) and done it visually (Figure 3).

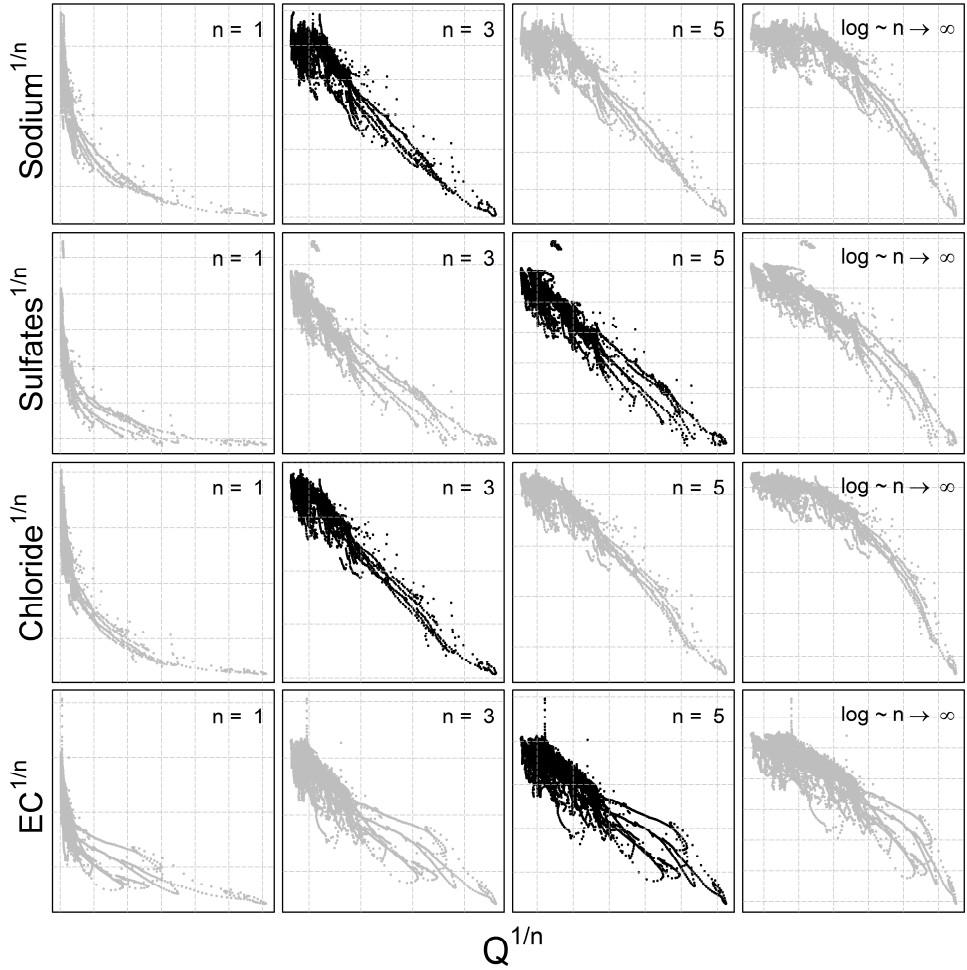


**Figure 3: C-Q behavior of three different chemical species and the electrical conductivity with**
**different 2S-APS transformations ($n$ =1, 3, 5, and log). The optimal power parameter (black dots)**
**was chosen based on the $R^2$ criterion. Note that we have removed the scale on the axes to focus**
**only on the change in shape in the C-Q relationship.**





**Table 2: Coefficient of determination ($R^2$) calculated for $n$ =1 (no transformation), $n$ = optimal value**
**for two-sided affine power scaling relationship (Figure 3) and n → ∞ (log–log space) for each ion**
**and for electrical conductivity (EC). Note that the $R^2$ is computed from transformed values.**

| Solute | $n$ | $R^2$ |
|---|---|---|
| Sodium | $n$ = 1 (no transformation) | 0.53 |
| | $n$ = 3 (optimal) | 0.73 |
| | $n$ → ∞ (log–log) | 0.53 |
| Sulfate | $n$ = 1 (no transformation) | 0.32 |
| | $n$ = 5 (optimal) | 0.81 |
| | $n$ → ∞ (log–log) | 0.77 |
| Chloride | $n$ = 1 (no transformation) | 0.52 |
| | $n$ = 3 (optimal) | 0.88 |
| | $n$ → ∞ (log–log) | 0.69 |
| EC | $n$ = 1 (no transformation) | 0.38 |
| | $n$ = 5 (optimal) | 0.79 |
| | $n$ → ∞ (log–log) | 0.74 |


The results given in Table 2 show the better quality of the fit obtained with the optimal value of $n$.

## 4. Numerical identification of the parameters for the 2S-APS relationship

The extremely large number of values in this high-frequency dataset may cause problems for a
robust identification over the full range of discharges using a simple linear regression. Indeed, the
largest discharge values are in small numbers (in our dataset only 1% of discharges are in the range
[2.6 $m^3s^{-1}$, 12.2 $m^3s^{-1}$], and they correspond to the lowest concentrations (see Figure 1)).
To address this question, we successively tested a large number of ($a$,$b$) pairs ($n$ remaining fixed at
the optimal value given in Table 2). Each pair yields a series of simulated concentrations ($C_{sim}$) that
can be compared with the observed concentrations ($C_{obs}$). Among the many numerical criteria that
could be used, we chose the bounded version of the Nash and Sutcliffe (1970) efficiency criterion
*NSEB* (Mathevet et al., 2006), which is commonly used in hydrological modeling. *NSEB* can be
computed on concentrations or on discharge-weighted concentrations (which corresponds to the
load). We chose the average of both, because we found that it allows more weight to be given to the
extremely low concentrations and thus to avoid the issue of under-representation of high-
discharge/low-concentration measurement points. Table 3 presents the formula for these numerical
criteria.
We retained as optimal the pair of ($a$,$b$) that yielded the highest $NSEB_{comb}$ value (we explored in a
systematic fashion the range [1–5] for $a$ and [-1.2–1.2] for $b$).





**Table 3: Numerical criteria used for optimization ($C_{obs}$ – observed concentration, $C_{sim}$ – simulated**
**concentration, $Q$ – observed discharge). The Nash and Sutcliffe (1970) efficiency (NSE) criterion is**
**well known and widely used in the field of hydrology. The rescaling proposed by Mathevet et al.**
**(2006) transforms NSE into NSEB, which varies between -1 and 1 (its optimal value). The advantage**
**of this rescaled version is to avoid the occurrence of large negative values (the original NSE**
**criterion varies in the range [-∞, 1]).**

$$NSE_{conc} = 1 - \frac{\sum_t (C_{obs}^t - C_{sim}^t)^2}{\sum_t (C_{obs}^t - \overline{C_{obs}})^2}$$ Eq. (4)

$$NSEB_{conc} = \frac{NSE_{conc}}{2 - NSE_{conc}}$$ Eq. (5)

$$NSE_{load} = 1 - \frac{\sum_t (Q^t C_{obs}^t - Q^t C_{sim}^t)^2}{\sum_t (Q^t C_{obs}^t - \overline{QC_{obs}})^2}$$ Eq. (6)

$$NSEB_{load} = \frac{NSE_{load}}{2 - NSE_{load}}$$ Eq. (7)

$$NSEB_{comb} = \frac{1}{2}(NSEB_{conc} + NSEB_{load})$$ Eq. (8)


## 5. Results

### 5.1 Results in calibration mode

The optimal values of $a$ and $b$ corresponding to the simulation of each ion and EC with the highest
$NSEB_{comb}$ criterion and the $n$ value identified in Figure 3 and Table 2 are presented in Table 4.
**Table 4: Summary of values $a$, $b$, and $n$ used to obtain the optimal $NSEB_{comb}$ criterion.**

| Ion | n | a | b | $NSEB_{comb}$ |
|---|---|---|---|---|
| Sodium | 3 | 2.70 | -0.60 | 0.68 |
| Sulfate | 5 | 2.20 | -0.55 | 0.69 |
| Chloride | 3 | 3.70 | -1.00 | 0.83 |
| EC | 5 | 4.20 | -0.70 | 0.77 |


For comparing the two relationships, we used the RMSE criterion. The results are shown in Table 5;
they illustrate (for our catchment) the better performance (i.e., lower RMSE value) of the proposed
2S-APS relationship for the three ions (sodium, sulfate, and chloride) over the power law
relationship. For EC, there is a slight advantage over the power law. A test of the equality of variance
(F-test) was performed between the RMSE obtained for the two relationships: Because of the very
large number of points in our dataset, all differences were highly significant (*p*-value <0.001)



**Table 5: Summary of values of RMSE criterion calculated for the three ions and EC.**

| Solute | 2S-APS | Power law |
|---|---|---|
| | RMSE | RMSE |
| Sodium | 1.00 mgL$^{-1}$ | 1.22 mgL$^{-1}$ |
| Sulfate | 2.17 mgL$^{-1}$ | 2.22 mgL$^{-1}$ |
| Chloride | 2.00 mgL$^{-1}$ | 2.91 mgL$^{-1}$ |
| EC | 42.0 μS.cm$^{-1}$ | 41.3 μS.cm$^{-1}$ |

Figure 4 illustrates the comparison of the quality of simulation over the entire calibration dataset between the power law and 2S-APS relationships. In general, the two-sided affine power scaling relationship yields better simulated concentrations than the classic power law relationship for the three ions (according to the results of Table 5). This is particularly evident over the low concentrations. This better performance is more apparent in the case of sodium and chloride ions.





176

Figure 4: **Comparison of simulated concentrations with observed concentrations for: (a) two-sided affine power scaling (2S-APS) relationship, (b) power law (calibration mode).**





### 5.2 Results in validation mode

For the validation mode, we applied the above-calibrated relationships to a different time period
(August 2017 to March 2018). The results are shown in Table 6. The RMSE criterion illustrates (for
our catchment) the better performance of the proposed 2S-APS relationship over the power law
relationship for all the solutes. Unlike the calibration case, the quality of the simulation of EC using
the 2S-APS relationship has a much better performance than the one simulated by the power law
relationship.
**Table 6: Summary of values of RMSE criterion calculated for the three ions and EC with the**
**validation dataset.**

| Solute | 2S-APS | Power law |
|---|---|---|
| | RMSE | RMSE |
| Sodium | 1.48 mgL$^{-1}$ | 1.90 mgL$^{-1}$ |
| Sulfate | 1.65 mgL$^{-1}$ | 2.33 mgL$^{-1}$ |
| Chloride | 3.69 mgL$^{-1}$ | 4.34 mgL$^{-1}$ |
| EC | 62.3 µS.cm$^{-1}$ | 78.8 µS.cm$^{-1}$ |


## 6. Conclusion

In this technical note, we tested and validated a three-parameter relationship (2S-APS) as an
alternative to the classic two-parameter one-sided power scaling relationship (commonly known as
"power law"), to represent the concentration–discharge relationship. We also proposed a way to
calibrate the 2S-APS relationship.
Our results (in calibration and validation mode) show that the 2S-APS relationship can be a valid
alternative to the power law: In our dataset, the concentrations simulated for sodium, sulfate, and
chloride as well as the EC are significantly better in validation mode, with a reduction in RMSE
ranging between 15 and 26%.

*Data availability*. Data will be available in a dedicated database website after a contract accepted on
behalf of all institutes.
*Competing interests*. The authors declare that they have no conflict of interest.
*Acknowledgments.* The first author acknowledges the Peruvian Scholarship Cienciactiva of CONCYTEC
for supporting his PhD study at Irstea and the Sorbonne University. The authors acknowledge the



EQUIPEX CRITEX program (grant no. ANR-11-EQPX-0011) for the data availability. We thank François
Bourgin for his kind review.

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

## 8. Appendix 1 – Description of the River Lab

In June 2015, the "River Lab" was deployed on the bank of the Avenelles River (within the limits of
the Oracle-Orgeval observatory, see Figure 5) to measure the concentration of all major dissolved
species at high frequency (Floury et al., 2017). The River Lab's concept is to "permanently" install a
series of laboratory instruments in the field in a confined bungalow next to the river. River Lab
performs a complete analysis every 30 min using two Dionex® ICS-2100 ionic chromatography (IC)
systems by continuous sampling and filtration of stream water. River Lab measures the concentration
of all major dissolved species ($[Mg^{2+}]$, $[K^+]$, $[Ca^{2+}]$, $[Na^+]$, $[Sr^{2+}]$, $[F^-]$, $[SO_4^{2-}]$ $[NO_3^-]$, $[Cl^-]$, $[PO_4^{3-}]$). In
addition, a set of physico-chemical probes is deployed to measure pH, conductivity, dissolved $O_2$,
dissolved organic carbon (DOC), turbidity, and temperature. The discharge is measured continuously
via a gauging station located at the River Lab site.
All the technical qualities, calibration of the equipment, comparison with laboratory measurements,
degree of accuracy, etc. have been well described in a publication by Floury et al. (2017).



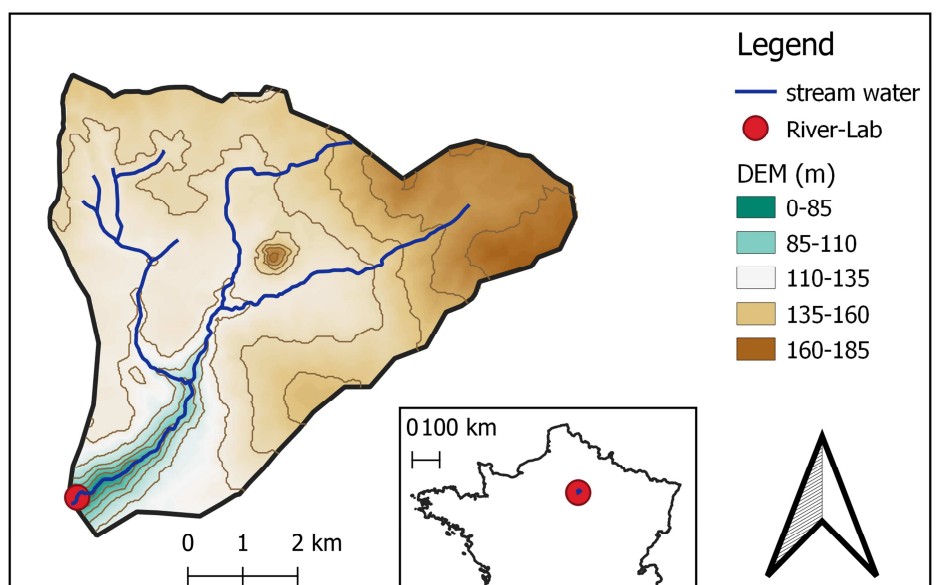


**Figure 5: Location of the River Lab (red dot) on the Avenelles River, Oracle-Orgeval observatory.**
