# Peer review of "Technical Note: 2 A two-sided affine power scaling relationship to represent the 3 concentration–discharge relationship"

_Hydrology and Earth System Sciences, 2019_

## Referee Comment (RC1) · Renata Romanowicz (Referee) · 15 Nov 2019

General comments

The Authors present a technical mote on the parameterisation of concentration-discharge relationships for an electrical conductivity and a number of different solutes, including sodium, sulphate and chloride. The classic one-sided power low parameterisation is substituted by a new, two-sided affine power parameterisation. The authors show that the new parameterisation scheme performs better as judged by the RMSE criterion. The paper is concise and well written. It would gain in scientific merit if the ranges of applicability of the new parameterisation and its predictive uncertainty were

discussed. The number of parameters is increased by one (from two to three), which might not seem to be much but it must introduce more uncertainty to the results.

Specific comments

1. Table 1: The authors use "sulfate" instead of sulphate throughout the whole paper. It is an American spelling and personally I would prefer the classic spelling.

2. Page 8 line 138 . . .(a,b) pairs from eq. 3 . . ..

―――――――――――――――――――

---

## Referee Comment (RC2) · Anonymous Referee #2 · 17 Dec 2019

General comments The manuscript on "A two-sided affine power scaling relationship to represent the concentration-discharge relationship" introduces an interesting alternative to conventional power law relationships. By using an additional degree of freedom the new approach is more flexible to relate concentration to discharge. It seems that better results compared to the conventional power law can be achieved especially when concentration variation is large. If my suggestions can be considered sufficiently I would like to support acceptance of the manuscript for publication.

Specific comments Line 50, Table 1: use mg S L-1 instead of Smg L-1, more over the coefficient of variation could be added to give a simple measure for the variation

within the data set, although the less often used (q90-q10)/$\mu$) statistic is already presented but without mentioning it in the manuscript anymore. Line 143 ff: the use of the NSEB criteria reduces the sensitivity of this objective function compared to the original NSE. If the concertation variability is small compared to the discharge variability solute loads are highly controlled by discharge. Therefor combining concentration with load objective function will further reduce the sensitivity of these criteria in those cases. To provide a most transparent evaluation I suggest to provide all five given criteria separately not only in the calibration mode but also in the validation mode. Line 169, Table 5: I would suggest to provide the mean concentration of the solutes in the table although they have been provided already in table 1 making the assessment of the RMSE easier Line 187, Table 6: here all five introduced evaluation criteria should be given to allow an assessment of the new approach in more detail, e.g. distinguish between concentration and load calculations. Line 187: It seems that the new approach has especially advantages if the variability of concentration and probably also discharge is large. If this is the case this would allow for a more detailed discussion of the advantages and possibly also limitations of the new approach.

---

## Author Comment (AC1) · 19 Dec 2019

Dear Colleague,

Thank you for your review, which will help us improve our technical note. Please find below a detailed response to the points you raised:

**It would gain in scientific merit if the ranges of applicability of the new parameterisation and its predictive uncertainty were discussed. The number of parameters is increased by one (from two to three), which might not seem to be much but it must introduce more uncertainty to the results.**
We have computed the predictive confidence interval, as it is common in case of linear regression (Jonnston, 1972 pp. 154-155; see also the discussion in Andréassian et al., 2007) and present the results in Figure 1 below. The figure speaks for itself: the predictive interval (blue surface for a 50 % predictive confidence interval, red for 95%) is much narrower for the 2S-APS relationship. This means that predictive uncertainty is more impacted by the error than by the number of parameters. We will add this graph in the final version of the paper.

[Figure]

**Figure 1: Predictive confidence interval computed for the 2S-APS relationship and the power-law for the 3 ions and the EC relationship. In blue the 50 % and in red the 95 % predictive confidence intervals**

**Specific comments**

**1. Table 1: The authors use "sulfate" instead of sulphate throughout the whole paper. It is an American spelling and personally I would prefer the classic spelling.**

We will change it in the revised manuscript

**2. Page 8 line 138... (a,b) pairs from eq. 3...**

We will change it in the revised manuscript

**References:**

Andréassian, V., Lerat, J., Loumagne, C., Mathevet, T., Michel, C., Oudin, L., and Perrin, C.: What is really undermining hydrologic science today?, Hydrological Processes, 21, 2819-2822, 10.1002/hyp.6854, 2007.

Jonnston, J.: Econometric Methods, McGraw - Hill Book Company, USA, 437 pp., 1972.

---

## Author Comment (AC2) · 19 Dec 2019

Dear Colleague,

Thank you for your review, which will help us improve our technical note. Please find below a detailed response to the points you raised:

**Specific comments**

**Line 50, Table 1: use mg S L-1 instead of Smg L-1, more over the coefficient of variation could be added to give a simple measure for the variation within the data set, although the less often used (q90-q10)/µ) statistic is already presented but without mentioning it in the manuscript anymore.**

With respect to the nomenclature of the sulphate ion, we will make the corresponding changes in the final manuscript.
We will replace (q90-q10)/µ) with the Coefficient of Variation (CV) in the final manuscript, and also add a few explanatory lines on this criterion.

**Line 143 ff: the use of the NSEB criteria reduces the sensitivity of this objective function compared to the original NSE. If the concertation variability is small compared to the discharge variability solute loads are highly controlled by discharge. Therefor combining concentration with load objective function will further reduce the sensitivity of these criteria in those cases. To provide a most transparent evaluation I suggest to provide all five given criteria separately not only in the calibration mode but also in the validation mode**
**Line 187, Table 6: here all five introduced evaluation criteria should be given to allow an assessment of the new approach in more detail, e.g. distinguish between concentration and load calculations.**

We will add the four remaining criteria (NSE and NSEB of concentration and load) in the final version of the manuscript.

**Line 169, Table 5: I would suggest to provide the mean concentration of the solutes in the table although they have been provided already in table 1 making the assessment of the RMSE easier**

We will add the mean concentration of solutes in Table 5 in the final version of the manuscript.

**Line 187: It seems that the new approach has especially advantages if the variability of concentration and probably also discharge is large. If this is the case this would allow for a more detailed discussion of the advantages and possibly also limitations of the new approach.**

We wanted to keep the discussion short because this is a technical note, however in the final manuscript we will add a short section showing the advantages and limitations of this new approach.